# Mean number of TeV muons in air showers measured with IceTop and IceCube

**Stef Verpoest[1]⋆ for the IceCube Collaboration†**

**1** Bartol Research Institute and Dept. of Physics and Astronomy, University of Delaware, Newark, DE 19716, USA

⋆ stef.verpoest@icecube.wisc.edu ,    † http://icecube.wisc.edu

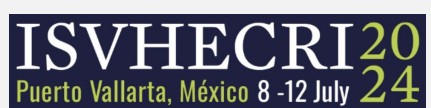

*22nd International Symposium on Very High Energy Cosmic Ray Interactions (ISVHECRI 2024) Puerto Vallarta, Mexico, 8-12 July 2024*

## Abstract

**IceCube is a cubic kilometer detector buried in the Antarctic ice at the South Pole. Combined with its surface component, IceTop, it constitutes a unique detector for air-shower physics in the PeV to EeV primary energy regime. In this contribution, a recent measurement of the mean multiplicity of muons with energies above several 100 GeV ("TeV muons") in near-vertical air showers seen in coincidence between IceTop and the IceCube in-ice array is reported. The results are consistent with expectations from simulations based on the hadronic interaction models used in the analysis: Sibyll 2.1, and the post-LHC models QGSJet-II.04 and EPOS-LHC. However, inconsistencies with other air-shower observables are found for all considered models. Notably, the observed density of GeV muons at large lateral distance in IceTop indicates a lighter cosmic-ray mass composition than the high-energy muon measurement.**

## 1 Introduction

Because of their low flux, cosmic rays with energies higher than about 100 TeV have to be studied indirectly by detecting the extensive air showers (EAS) produced when they interact in the Earth's atmosphere. The interpretation of the measurements requires detailed simulations of the development of the EAS. Several experiments have, however, reported discrepancies between the muon number predicted in simulations and observations [1]. This issue, known as the Muon Puzzle, is attributed to an incomplete description of the high-energy hadronic interactions in the shower [2]. It prevents an accurate determination of the cosmic-ray mass composition, which is important for identifying the sources of cosmic rays, both by itself and for the calculation of atmospheric backgrounds for neutrino and gamma-ray observatories.

With its combination of a surface air-shower array and a deep in-ice detector, the IceCube Neutrino Observatory [3] can perform unique tests of muon production in EAS. The surface

13 detector, IceTop, measures low-energy muons at large distances from the shower axis [4].
14 Only muons with energies greater than several hundred GeV, referred to as "TeV muons", can
15 reach the in-ice detector. In this contribution, an update on Refs. [5] and [6], we present
16 a measurement of the average multiplicity of muons with $E_\mu > 500$ GeV in near-vertical air
17 showers as a function of cosmic-ray energy between 2.5 and 100 PeV. In Fig. 1, predictions
18 obtained from CORSIKA simulations [7] are shown.

## 2   Cosmic rays with IceCube

20 The IceCube Neutrino Observatory is located at
21 the geographical South Pole. Its surface compo-
22 nent, IceTop, has an elevation of about 2.8 km
23 a.s.l., corresponding to an average atmospheric
24 depth of about ∼690 g/cm$^2$ [8]. It consists of 81
25 stations of two ice-Cherenkov tanks, deployed on
26 a triangular grid with a 125 m spacing. Each tank
27 has two Digital Optical Modules (DOMs) to de-
28 tect Cherenkov light produced by shower parti-
29 cles. The IceCube in-ice array consists of about
30 5000 DOMs deployed on vertical strings between
31 1450 m and 2450 m below the surface [3] below
32 the surface. The strings follow approximately the
33 same grid as the surface stations, instrumenting a
34 total volume of about 1 km$^3$.

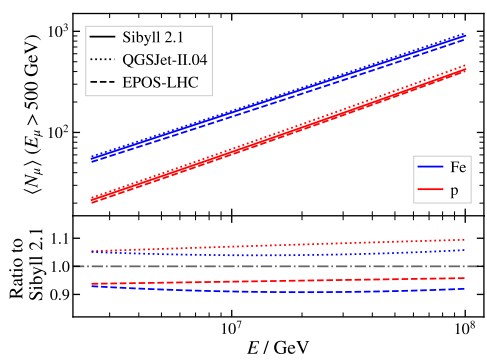

Figure 1: Mean number of muons above 500 GeV in EAS with $\cos\theta > 0.95$ obtained from simulations.

35  IceTop detects EAS from cosmic rays in the PeV to EeV range. As a result of its high altitude,
36 vertical EAS are detected near the shower maximum, and IceTop signals are dominated by
37 the electromagnetic component, except at large distances from the shower axis where muons
38 become more prominent. Muons with energies over several hundred GeV, which are typically
39 closely aligned with the shower axis, can propagate all the way to the in-ice detector and be
40 observed in coincidence with the IceTop signals.

## 3   High-energy muon number analysis

42 The analysis uses events whose shower core is contained in IceTop and which have a coincident
43 muon bundle in the in-ice array. The zenith angle is limited to $\cos\theta > 0.95$ or $\theta \lesssim 18°$. Sev-
44 eral existing reconstructions are applied to the events. A standard air-shower reconstruction is
45 performed using only information from IceTop, reconstructing the shower core, direction, and
46 the shower size $S_{125}$ [8]. This is done by fitting the observed signals with a lateral distribution
47 function (LDF) and shower front model, respectively. The in-ice signals corresponding to the
48 high-energy muon bundle are used to perform an energy-loss reconstruction. The reconstruc-
49 tion algorithm fits the energy deposited by the muon bundle in track segments of 20 m along
50 its path [9], for which the shower axis from the IceTop reconstruction is used.

51  Quality cuts developed for earlier analyses are applied [10]. They result in a sample of
52 events with a well-reconstructed direction and shower size, with the shower core contained
53 within the boundaries of the IceTop array. The in-ice selection ensures that a high-energy
54 muon signal with a successful energy-loss reconstruction is present.

## 3.1    Neural network reconstructions

A neural network (NN) is used to reconstruct two quantities for every event: the primary cosmic-ray energy, $E$, and the number of muons with $E_\mu > 500\,\text{GeV}$ in the shower at the surface, $N_\mu$. The segmented energy-loss reconstruction of the muon bundle in the in-ice detector forms the input to a recurrent NN layer, a typical choice for sequential data. The corresponding output is combined with the reconstructed $S_{125}$ and $\theta$, feeding into a number of fully-connected layers which return both $E$ and $N_\mu$.

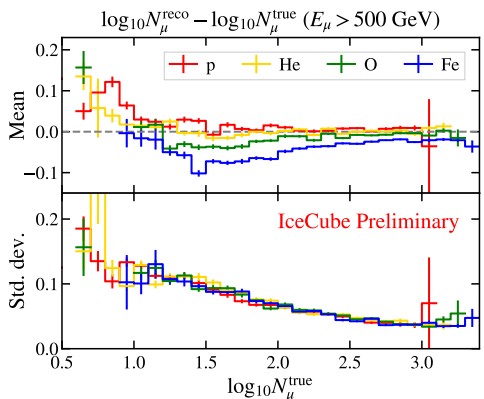

Figure 2: Bias (top) and resolution (bottom) of the $N_\mu$ reconstruction of the NN.

   To train the neural network, Monte Carlo simulations using Sibyll 2.1 with four primary nuclei (p, He, O, Fe) were employed. The atmospheric model is an average South Pole April atmosphere, a good approximation of the yearly average atmosphere [11]. The bias and resolution of the NN reconstruction of $N_\mu$ are shown in Fig. 2. (The energy reconstruction performance can be seen in Ref. [6].)

## 3.2    Monte Carlo corrections

The average muon multiplicity $\langle N_\mu \rangle$ as function of primary energy is estimated by binning events in the NN-reconstructed $E$ and calculating the mean reconstructed $N_\mu$. This is compared to the true $N_\mu$ binned in true $E$ in simulation. The ratio of reconstructed and true $\langle N_\mu \rangle$ is shown in Fig. 3 (left), where a mass-dependent bias is visible. These ratios are fitted with quadratic functions, which are used as correction factors to unbias the results obtained from data.

   An iterative correction procedure is used to take the mass dependence of the correction factors into account without assuming a specific mass composition. It exploits the approximately linear dependence of the correction factors on $\ln A$, where $A$ is the nuclear mass, which can be seen in Fig. 3 (left). It also uses the fact that a muon measurement is in itself a measure of the mass composition. The average $\langle N_\mu \rangle$ is compared to predictions from simulations of proton and iron showers using

$$z = \frac{\ln\langle N_\mu \rangle - \ln\langle N_\mu \rangle_\text{p}}{\ln\langle N_\mu \rangle_\text{Fe} - \ln\langle N_\mu \rangle_\text{p}}, \tag{1}$$

also known as the "$z$-scale" [1]. The Heitler-Matthews model predicts that this corresponds to an estimate of the mass composition as $z \approx \ln A / \ln 56$ [12]. The correction factor corresponding to the reconstructed $z$ is estimated by linearly interpolating between the correction factors for proton and iron from Fig. 3 (left) in $\ln A$. The resulting correction factor is then applied to the initial $\langle N_\mu \rangle$. The updated estimate of $\langle N_\mu \rangle$ is used to construct a new correction factor. This process can be repeated until $\langle N_\mu \rangle$ values converge. Fig. 3 (right) shows an example of this correction procedure using simulations. The approach successfully recovers $\langle N_\mu \rangle$ values consistent with the true values, independent of the mass composition.

   Correction factors were likewise derived based on QGSJet-II.04 and EPOS-LHC simulations. With these, results can be derived from experimental data under different model assumptions, as the correction factors effectively replace the dependence of the results on the model used in NN-training by a dependence on the model used to derive the correction factors.

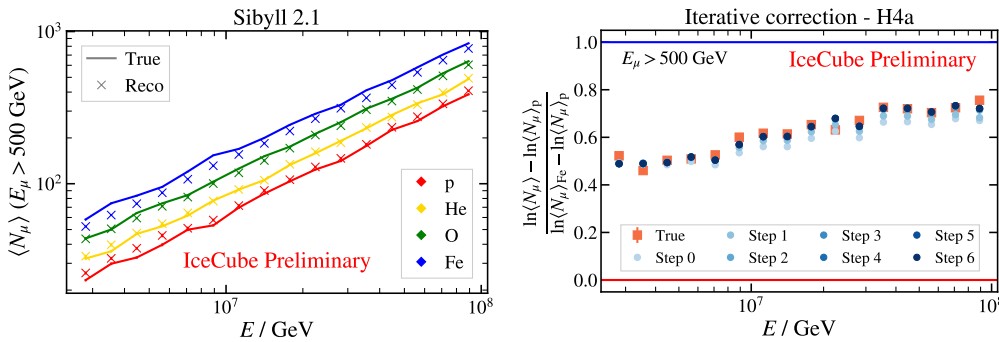

Figure 3: Left: Ratio of $\langle N_\mu \rangle$ using reconstructed and true values for $E$ and $N_\mu$ in simulation. The quadratic fits are used as correction factors in further analysis. Right: Example of the iterative correction applied to simulations weighted to the H4a flux model. After several steps, convergence to values consistent with the simulated true values is observed.

### 3.3  Results and discussion

The analysis uses one year of data, May 2012 to May 2013. The event selection reaches full efficiency for primary cosmic rays with energies above 2.5 PeV, which is chosen as the low-energy bound of the analysis. The analysis extends up to 100 PeV.

Fig. 4 shows $\langle N_\mu \rangle$ derived from the experimental data using the hadronic models Sibyll 2.1, QGSJet-II.04, and EPOS-LHC for the correction factors. Also shown are predictions for proton and iron showers from the corresponding model. The results are also plotted as $z$-values by scaling them according to the proton and iron expectations, following Eq. (1). They are shown together with expectations from the cosmic-ray flux models GSF [13], GST [14], and H3a [15]. All results are close to the model predictions, with the results based on EPOS-LHC implying a slightly heavier composition than those from Sibyll 2.1 and QGSJet-II.04.

The systematic detector uncertainties included are the same as those described in Ref. [10], but with a more conservative value of 10% for the uncertainty of the DOM efficiency. The dominant uncertainty is related to the modeling of the South Pole ice in simulation. Smaller uncertainties related to IceTop snow correction and charge calibration are also included.

The high-energy muon measurement presented here can be compared to a measurement of the density of low-energy ("GeV") muons at the surface performed with IceTop [4]. If the simulations correctly describe the EAS development, the $z$-values for both measurements, GeV and TeV muons, should be consistent. We find this to be the case for the values obtained based on Sibyll 2.1. However, for the post-LHC models QGSJet-II.04 and EPOS-LHC a tension is observed as a result of the larger number of low-energy muons compared to Sibyll 2.1. We note that preliminary IceCube results also indicate a discrepancy between the Sibyll 2.1 muon results and a shower observable related to the slope of the IceTop LDF [16].

## 4  Conclusion and outlook

We have presented a measurement of the mean number of muons above 500 GeV in near-vertical EAS for primary cosmic-ray energies between 2.5 PeV and 100 PeV, using the surface and in-ice detectors of IceCube. The results are consistent with predictions from simulations for all hadronic models included in the analysis, i.e. Sibyll 2.1, EPOS-LHC, and QGSJet-II.04.

The results have also been compared with an earlier measurement of muons at the surface with IceTop [4]. Inconsistencies between the measurements are observed for QGSJet-II.04 and EPOS-LHC, indicating that these models do not describe the experimental data consis-

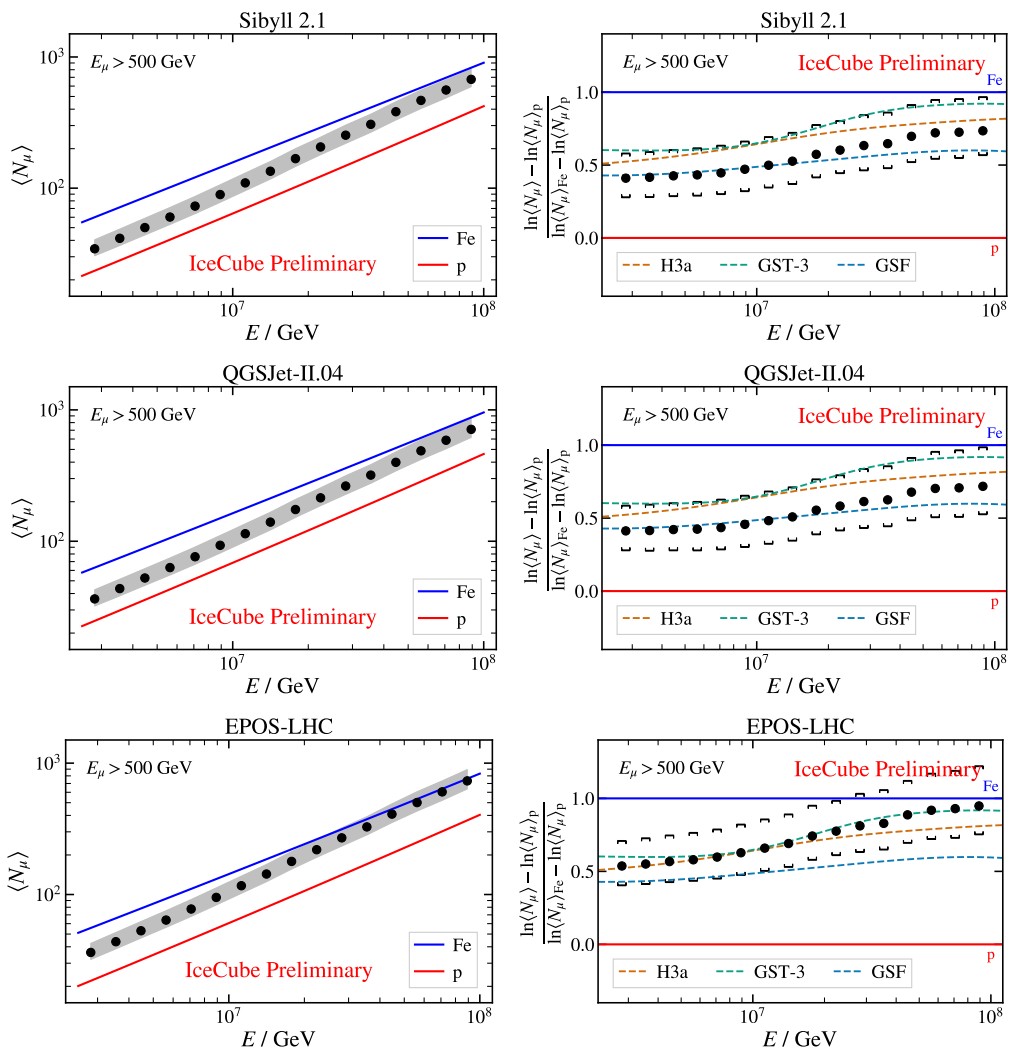

Figure 4: Mean number of muons with $E_\mu > 500\,\text{GeV}$ in near-vertical EAS obtained from data assuming different hadronic models. The right column shows the z-values corresponding to the results (Eq. (1)), together with expectations for different composition models. The bands (left) and brackets (right) indicate the systematic uncertainty, statistical uncertainties are too small to be visible.

tently. In addition, preliminary work indicates the existence of inconsistencies with a different observable for Sibyll 2.1 [16].

In the future, we plan to decrease the systematic uncertainties related to the ice model and extend the analysis to higher primary energies, as well as to include Sibyll 2.3d simulation [17]. With improved methods to determine the low-energy muon content in EAS (e.g. as in Ref. [18]), the spectral information obtained from muon measurements with IceCube will provide unique constraints on hadronic interactions in EAS.

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
