# Peer review of "Mean number of TeV muons in air showers measured with IceTop and IceCube"

_SciPost Physics Proceedings_

## Round 1 · Referee Report · Anonymous (Referee 1) · 2025-1-11

Strengths

1) The topic is valuable and is an open point in EAS studies 2) The analysis is new and it compares with past results 3) Results are interesting and useful for the full community and for the refinement of the hadronic interaction models

Weaknesses

1) Section 3.2 is hard to follow because of a mismatch between the figure 3left and what was supposed to be plotted there 2) More in general the limit to the number of pages which is probably imposed by the length required for the proceedings makes less clear the paper. As an example interesting results such as the behaviour of the models compared with iceTop data alone can be found only opening the papers in the references. It would have been nice to have all the results in a figure within this proceedings but clearly there is not enough space for it.

Report

The paper describes the analysis and shows the results of the mean number of TeV muons detected by Icecube in coincidence with IceTop. The results are extremely interesting because allow to assess information difficult to obtain with EAS arrays which can access only the muon component of low energy. The paper in general reads well, the results are well described, however, I found difficulties in understanding section 3.2 due to the wrong assignment of the plot presented on fig.3 left. Therefore, some assumptions have to be taken as granted.

Requested changes

1) lines 31-32: remove 'below the surface' which is repeated twice in the text. 2) lines 77-78: fig. 3left does not show any correction ratio. The plot should be changed according to the text. 3) lines 81-83: same as 77-78. The sentence is not directly supported by the plot. 4) Figure 3 left and it's caption do not match.

Recommendation

Ask for minor revision

  • validity: good
  • significance: high
  • originality: high
  • clarity: ok
  • formatting: excellent
  • grammar: excellent

Author:  Stef Verpoest  on 2025-02-17  [id 5224]

(in reply to Report 1 on 2025-01-11)

I thank the referee for their detailed examination of the text.

The proceedings have been updated with all the requested changes, most importantly the identified inconsistency between the text and Fig. 3 (left).

The figure requested by the referee comparing the low- and high-energy muon results from IceCube, which is not included in the article due to the page limit, will be included in a future journal publication.

---

## Editorial Decision

accepted_in_target_journal